# The Effect of Substrate on the Nutrient Content and Fatty Acid Composition of Edible Insects

**DOI:** 10.3390/insects13070590

**Published:** 2022-06-28

**Authors:** Kati Riekkinen, Kati Väkeväinen, Jenni Korhonen

**Affiliations:** Institute of Public Health and Clinical Nutrition, School of Medicine, Faculty of Health Sciences, University of Eastern Finland, 70210 Kuopio, Finland; kati.vakevainen@uef.fi (K.V.); jenni.korhonen@uef.fi (J.K.)

**Keywords:** insects as food, nutritional quality, essential fatty acids

## Abstract

**Simple Summary:**

The search of new sustainably produced protein sources for food and feed is vigorously under investigation. One promising possibility is to increase the use of edible insects as a part of our diet. The nutrient content of edible insects, in particular, a high content of good quality protein and unsaturated fatty acids with essential fatty acids, is an important health aspect when screening the most potential insect species for human consumption. Based on this review, the substrate affects the nutrient content of insects. Moreover, our correlation calculations demonstrated that the fatty acid content of the substrate influences the nutritional value of insects. In general, high content of unsaturated fatty acids in the substrate increased the amount of unsaturated fatty acids in insects. For example, the content of essential fatty acids, linoleic and alpha linolenic acids, can be raised by feeding insects with modified substrate. Thus, edible insects can be a healthy protein source to ease the increased demand for high quality food.

**Abstract:**

Demand for new food sources and production methods is increasing due to overall population growth, as well as the aim towards more sustainable use of natural resources and circular economy. Edible insects already used in many parts of the world have recently attracted interest as a new protein source in Europe, and novel food acceptance procedures are ongoing in the European Union for several insect species. In this paper, the effects of substate on the nutritional value, especially the fatty acid composition, of edible insects were reviewed and correlation calculations performed. The nutritional value of edible insects is an important health aspect, in particular, a high content of good-quality protein and unsaturated fatty acids with essential fatty acids, and an optimal fatty acid n6/n3 ratio. On the basis of our findings, the nutrient content of insects can be modified by using a feed substrate carefully designed for each individual insect species. In addition, our correlation calculations demonstrated that the contents of linoleic and alpha linolenic acids in insects reflected the contents of these acids in the substrate. In conclusion, optimizing the composition and structure of the substrate and rearing conditions and duration for each insect species might also aid standardization of the nutritional composition of edible insects.

## 1. Introduction

Insects have been a part of the human diet for a long time, especially in Asia, Africa, and South America [1]. Globally, the demand for food is increasing due to population growth, and thus traditionally produced animal protein will not be sufficient for the whole population. Therefore, edible insects could serve as an alternative protein source. In addition, insect rearing is considered to be a more sustainable production method than, e.g., traditional cattle farming, because insects consume fewer natural resources, such as water and land, and cause fewer environmental emissions [2,3].

The use of insects as food has increased in Europe in recent years [4]. Crickets, grasshoppers, and mealworms are amongst the insects most commonly produced as food, and the main commercial products are flavored snacks, energy bars, and powders used as sports supplements [4]. Insects and insect-derived products are categorized as novel foods within the European Union (EU), and the Novel Foods Regulation (EU) 2015/2283 guidelines are applied during manufacturing of insect products [5]. House cricket (*Acheta domesticus*), yellow mealworm (*Tenebrio molitor* larvae), lesser mealworm (*Alphitobius diaperinus* larvae), European honeybee (*Apis mellifera* male pupae), black soldier fly (*Hermetia illucens* larvae), and migratory locust (*Locusta migratoria*) are currently permitted for use as food in Finland [6]. However, the transitional legal phase is in progress, and the European Commission will take the final decision on insects approved for consumption as a novel food when the safety assessments have been completed.

Commonly used substrates for rearing edible insects are chicken feed and soybean meal, but new options are continuously tested. For example, side streams or byproducts from the food industry could offer economical and sustainable ways to produce animal protein. When side streams or byproducts are used as substrates, feed legislation must carefully be taken into consideration and the safety and quality of substrates ensured. The EU legislation (EU/2017/1017) defines substrates that can be used for the rearing of edible insects in Europe and, for example, manure, catering waste, and certain other wastes are not permitted substrates [7].

The nutritional composition of substrate is an important factor for the efficient growth of insects. For this reason, it would be beneficial to formulate the most suitable substrate for each reared insect species. Crucial aspects of optimization of the substrate include, for example, protein and fat contents, which should not be either too high or too low. According to Van Broekhoven et al. [8], low protein level in the feed substrate delayed the growth of mealworms, whereas a high protein level increased uric acid secretion. Similarly, too-high fat content in the substrate caused death of yellow mealworms in the adult stage, probably due to substrate agglomeration interfering with breathing and movement of insects [9].

Protein and fat contents of insects affect their nutritional value, and, in particular, a high content of essential amino acids and mono- and polyunsaturated fatty acids with a good fatty acid profile increases the nutritional quality. The protein content of insects varies according to species, age, rearing substrate, and analysis method [10]. Payne et al. [11] collected data of the nutrient composition of twelve edible insects and observed high standard deviations within the results, over 50% of the mean. For example, the protein contents of house cricket (*Acheta domesticus*) and yellow mealworm (*Tenebrio molitor*) were 7.5–23.7% and 11.4–30.4% of fresh weight, and fat contents 2.4–6.7% and 6.4–23.0%, respectively [11].

The total amount of crude protein of food or feed is often assayed by the Kjeldahl method. With this method, the amount of nitrogen is determined and converted to crude protein by the nitrogen-to-protein conversion factor (generally 6.25). According to Jonas-Levi and Martinez [12], the protein content of insects has been overestimated due to the presence of non-digestible chitin, which is a polysaccharide composed of units of *N*-acetylglucosamine containing a nitrogen atom. This overestimation can be avoided by analyzing non-digestible chitin separately. Alternatively, for example, Ritvanen et al. [13] proposed the use of a nitrogen conversion factor of 5.0 for house cricket and field cricket.

The nutritional composition of edible insects has been surveyed in several review publications [14,15,16]. By contrast, the effects of rearing substrate on edible insects have been less well reviewed. The aim of this review was to summarize the results of (1) macronutrients and minerals and (2) fatty acid composition of edible insects (currently permitted for use as food in Finland) and used substrates, and to calculate (3) the correlations between the levels of selected fats, namely, linoleic and alpha linoleic acid, in edible insects and substrates. The calculations were performed to evaluate how efficiently these fatty acids can be transferred from substrate to the insects.

## 2. The Effect of Substrate on the Macronutrient and Mineral Composition of Edible Insects

In the following sections, the main findings of the effect of substrate on the macronutrient and mineral composition of edible insects are summarized. Appendix A show, in detail, the nutritional data of the substrates used for rearing of edible insects, and Appendix A presents the nutritional data of the edible insect species considered in this review. To the best of our knowledge, the effect of substrate vitamins on the chemical composition of insects has not been studied comprehensively. Hence, vitamins are not addressed within this review.

### 2.1. House Cricket (Acheta domesticus)

Substrates with different protein and fat contents have been used to grow crickets. The total crude protein of substrates for rearing house crickets has been reported as 6.3–22.9% and after rearing, the protein of crickets has been 48.1–76.2% of dry matter [17,18]. It appears that high protein content of the substrate could increase the protein content of crickets, but protein intake of crickets is also affected by other factors, such as vitamins and minerals, and composition and structure of the substrate [17]. After being fed commercial protein-rich (22%) substrate (Pure pride cricket feed), crickets were detected to be, on average, 9% heavier (body weight) and longer (length) than crickets fed commercial low-protein (16%) substrate (Betagro chicken feed) [17]. Furthermore, when the insect can utilize protein more effectively as an energy source, due to, e.g., good digestibility of protein, the protein-rich substrate may also decrease the fat content of the crickets [17]. By contrast, protein-rich (22%) substrate added with fresh pumpkin pulp with a high carbohydrate content increased fat content in the crickets [17]. According to Oonincx et al. [18], different fat contents in substrates were not seen to correspond with the amounts of total fatty acids (TFA) in the crickets.

Dietary calcium and phosphorus levels and their ratio are important for normal growth and development of insects [19]. In a study by Bawa et al. [17], the mineral content of the house cricket was observed to be influenced by the substrate, and a high protein content of substrate increased not only the protein but also the amounts of sodium, calcium, phosphorus, and potassium in the crickets. Thus, nutrient combination and balance of the diet, not only high calcium concentration of the substrate, increased calcium intake of the crickets [19]. The palatability of substrate can be increased by, e.g., fatty and amino acids, and the acceptance by a pleasant texture obtained by grinding or crushing [19]. Enhanced palatability and acceptance increased consumption of substrate, which, in turn, also increased the yield of the crickets [19].

### 2.2. Black Soldier Fly (Hermetia illucens)

Black soldier fly (BSF) has been extensively studied due to its efficient utilization of different kinds of substrates. In addition, it has a fast growth rate and life cycle. BSF can be fed with organic byproducts of the agri-food industry, and thus the environmental impact of the produced protein is lower than if a special substrate, such as soybean meal, is used for rearing [20]. In a study by Bava et al. [20], BSF larvae fed with high-non-fibrous-carbohydrate (NFC) content substrates (hen diet, maize distiller) had higher final weight and a faster prepupal stage than larvae fed with low NFC content substrates (okara, brewer’s grain). High crude protein and crude fat (ether extract, EE) contents in substrates were observed to reflect higher proteins and fat content of larvae [20]. By contrast, Danieli et al. [21] observed that protein-rich substrate named TMD3, containing ca. 14% protein, did not increase the protein level in BSF prepupae, but high NFC content in the substrate named TMD1, containing ca. 69% NFC, increased the fat content in prepupae by approximately 50%, although the total fat amount was the same in all tested substrates. The protein-to-carbohydrate ratio of substrate was assumed to affect the protein-to-fat content ratio in BSF [21]. According to Beniers and Graham [22], high protein content in the substrate increased the weight of BSF larvae, as well as the content of protein and fat. However, the percentage of substrate protein had a significant (*p* = 0.000) negative correlation with the percentage of larvae protein, while the percentage of substrate protein had a positive significant (*p* = 0.000) correlation with the percentage of larval fat [22]. By contrast, substrate carbohydrate was not seen to have a significant effect on the fat content of BSF larvae [22].

In a study by Jucker et al. [23], BSFs were reared on three different substrates: fruits (F), vegetables (V), and a mixture of fruits and vegetables (FV). The highest crude protein content was in vegetables and the lowest in fruits, whereas the highest carbohydrate content was in fruits and the lowest in vegetables [23]. The chemical analysis revealed that BSF prepupae reared on substrate F had the highest fat content and the lowest crude protein content, and prepupae reared on substrate V had the lowest fat content and on substrate FV the highest crude protein content, respectively [23]. In a study by Meneguz et al. [24], BSF larvae reared on protein- and fat-rich brewery byproduct (BRE) had the highest protein content compared to larvae reared on winery byproduct (WIN) or fruit (FRU) or a mixture of fruit and vegetables (VEGFRU). Larvae reared on BRE also had good efficiency of conversion of digested food (ECD) value, a high total final biomass production, and a short developmental period [24].

Liu et al. [25] observed that the chemical composition of BSF larvae changed due to differences in substrates, and larvae fed on the substrate named as standard diet had the highest crude protein and fat content. The substrate standard diet had high NFC content but not the highest protein content, compared with other substrates, and thus the chemical composition of larvae was considered to be a result of overall composition of the substrate [25]. In addition, the chemical composition changed during larval development; in particular, crude protein content was observed to decrease, and crude fat content to increase during the development [25]. According to Liu et al. [25], the most suitable nutritional ratio of substrate for BSF larvae is protein:fat:digestible carbohydrate 2:1:2. In a study by Galassi et al. [26], it was seen that ash (*p* = 0.008) and fat (EE) (*p* = 0.040) contents of BSF larvae can be affected by the growth substrate. Furthermore, a positive correlation was estimated to hold between chitin content of larvae and neutral detergent fiber of the substrate [26]. By contrast, substrate protein was not observed to have an effect on crude protein content of larvae [26].

Crude protein content in high protein–low fat substrate (HPLF) was 22.9%, and in high protein–high fat substrate (HPHF), 21.9% [18]. After rearing, the protein content of BSF larvae was 43.5% of dry matter when fed on HPLF and 46.3% of dry matter when fed on HPHF [18]. Larvae reared on low-protein substrates (low protein–high fat (LPHF) and low protein–low fat (LPLF)) had ca. 38% of dry matter of protein, and thus the protein content of BSFs was observed to increase due to the protein content of the substrate [18]. The fat content in substrate HPHF was 9.5 times higher than in substrate HPLF, but the difference in fat contents between BSF larvae reared on HPLF or HPHF was only 0.8% [18]. According to Oonincx et al. [18], the nutrient content of BSF larvae showed only minor changes due to varying substrate composition. By contrast, Shumo et al. [27] observed in their study that protein, fat, minerals, amino acids, and fiber content of BSF larvae can be affected by the choice of rearing substrate. However, the crude protein content was not the highest in larvae reared on the substrate with the highest protein content, and a similar finding was also seen for calcium content [27].

The protein content of BSF prepupae was not observed to differ greatly between prepupae reared on various substrates, despite great differences in protein content of the substrates [28]. By contrast, fat (EE) and ash contents of BSF were influenced by substrate, and a high correlation (*p* = 0.023) was observed in ash content between substrates and prepupae, as well as between fat (EE) content (*p* = 0.030) of the prepupae and NFC content of the substrate [28]. No correlations were observed in mineral levels, such as calcium, between substrates and prepupae [28]. However, in a study by Adebayo et al. [29], the crude protein content of BSF prepupae mainly reflected the nutrient content of the substrate, although prepupae fed on a substrate named food remains (FR), with the highest protein content, did not have the highest protein content. By contrast, fat (EE) content of the prepupae reflected more the fat content of the used substrate [29]. Interestingly, the ash content in substrate named chicken feed (CF) was the highest, although prepupae fed on CF had the lowest ash content [29]. However, the calcium content of prepupae followed the calcium content of the substrate [29].

In a study by Tschirner and Simon [30], three different substrates, a mixture of middlings, dried distillers’ grains with solubles, and dried sugar beet pulp, were used for rearing BSF larvae. The control group of larvae was fed by mixture of middlings, the protein group of larvae was fed by dried distillers’ grains with solubles, and the fiber group was fed by dried sugar beet pulp [30]. The substrate dried distillers’ grains with solubles had the highest crude protein and fat (EE) contents [30]. The substrate dried sugar beet pulp had twice as much fiber as dried distillers’ grains [30]. In addition, the substrate dried sugar beet pulp had low protein and very low fat (EE) content compared to other tested substrates [30]. After rearing, larvae of fiber group had the highest protein and ash contents and the lowest fat (EE) content [30]. On the contrary, the highest fat (EE) content had the larvae of the protein group [30]. Calcium and phosphorus concentrations were the highest in the fiber group larvae but none of the tested groups had favorable Ca:P ratio [30]. However, according to Tschirner and Simon [30], the nutrient composition of BSF larvae can be affected by feeding substrate but, for example, too-high protein content in substrate could inhibit the development of larvae and lead to lower total yield [30].

### 2.3. Yellow Mealworm (Tenebrio molitor L.)

The chemical composition of yellow mealworm was observed to be affected by the substrate [31]. Larvae fed on brewery spent grains (SG) had higher crude protein and carbohydrate contents than larvae reared on bread (B) or cookies (C) [31]. The crude protein content of SG was higher than that of the other substrates, and crude fat (EE) content was the highest in cookies, whereas bread had the highest carbohydrate content [31]. The highest fat content was observed in larvae fed on bread and cookies, although bread had the lowest fat content [31]. Melis et al. [32] reported that rearing mealworms on dried brewer’s spent grains (BSG) led to a higher feed conversion ratio and efficiency in conversion of ingested food than conventional wheat bran (WB). The crude fat content was considerably lower in mealworms reared on BSG than in mealworms reared on WB, even though the fat content in BSG was almost twofold higher than in WB [32]. This was assumed to be due to the high fiber content (cellulose and lignin) of BSG [32]. The protein content in yellow mealworms (54% of dry matter) reared on high-protein substrates, HPHF and HPLF, was higher than in mealworms (44–48% of dry matter) reared on low-protein substrates, LPHF and LPLF [18]. The fat (TFA) content in mealworms was more variable due to the substrate, although the substrate fat level did not consistently reflect the amount of fat in insects [18].

The use of novel biomasses as a substrate to increase the nutritional quality of yellow mealworms has been surveyed, and one option could be pulp flour of bocaiuva (*Acrocomia aculeata* (Jacq.) Lodd) palm tree [9]. In the study by Alves et al. [9], yellow mealworms were fed four different substrates: the control substrate A, containing 50% wheat flour and 50% soybean flour; control substrate mixed with 50% of bocaiuva pulp flour (B); control substrate mixed with 50% of ground bocaiuva kernel (C); and 50% ground bocaiuva pulp flour and 50% bocaiuva kernel (D) [9]. The control substrate A had higher protein and carbohydrate contents and lower fat content compared to the other substrates [9]. The fat content of mealworms reared on substrate B compared to mealworms reared on control substrate A was not significantly different, but the protein and fiber contents were higher in mealworms fed on the control substrate A [9]. Increasing the amount of bocaiuva pulp flour or ground bocaiuva kernel in the substrate also increased its lipid level, and thus the fat content of substrate C was 29.3% and of substrate D 36.9% [9]. In the study, mealworms reared on substrates C and D died after 130 days, probably due to the high lipid content of the substrates [9]. However, according to Alves et al. [9], bocaiuva pulp flour could be a promising alternative as a component of substrate for growing mealworms without risking the nutritional quality.

Olive pomace is a byproduct from the olive oil industry, and it has been assessed as a substrate (S3–S5) for yellow mealworm larvae [33]. Substrate S3 contained 25% olive pomace, and it had the highest protein and carbohydrate (nitrogen free extract, NFE) contents but the lowest fat and fiber contents compared to other substrates containing olive pomace [33]. Mealworm larvae reared on substrate S3 had the highest protein content, whereas fat content was the highest in larvae fed on substrate S5 [33]. Substrate S5 had the highest amount of olive pomace (75%), and thus had the highest fat content, but the fat contents were not significantly different between mealworm larvae reared on substrates with different olive pomace contents [33].

Yellow mealworms have also been grown on substrates originating from organic byproducts of beer brewing, bread/cookie baking, potato processing, and bioethanol production [8]. High protein–low starch substrate (HPLS) had the highest crude protein (39.1%) and crude fat content (5.8%) [8]. After rearing, the protein or fat contents of mealworms did not differ greatly based on substrate, except that the fat content of the mealworms fed on low protein–high starch (LPHS) was lower than that of mealworms fed other substrates [8]. According to Van Broekhoven et al. [8], yellow mealworms are able to secrete excess protein and to adjust the amount of protein in their body to the optimal level. By contrast, the fat content of substrate had an influence on the fat content of mealworms, but low nutritional quality of substrate LPHS was also assessed to lead to lower fat content [8].

In a study by Zhang et al. [34], three byproducts, mushroom spent corn stover (MSCS), highly denatured soybean meal (HDSM), and spirit distillers’ grains (SDG), were used as substrates to rear yellow mealworms. The substrate MSCS had the lowest protein content, and HDSM had the highest protein and the lowest fat content, whereas the substrate MSCS had the highest fat and carbohydrate contents [34]. All mealworms reared on different byproducts had high protein content (70.1–76.3% of dry matter) [34]. Mealworms fed on MSCS had the lowest fat content, which was assessed to be due to undigestible fibers in MSCS [34]. Zhang et al. [34] observed that mealworms fed on fiber-rich substrates had higher protein and lower fat contents than mealworms fed on fiber-free substrates.

Calcium level of yellow mealworm was observed to increase by substrate with high calcium content supplement while phosphorus did not have a similar effect [19]. The palatability and other nutrients of the substrate could also increase the consumption of the substrate, and thus the availability of calcium [19]. In a study by Oonincx et al. [18], the phosphorus level of the substrate had no influence on phosphorus content of mealworm, but phosphorus concentration and crude protein content were seen to correlate significantly (*p* = 0.001) in mealworms.

### 2.4. Lesser Mealworm (Alphitobius diaperinus)

High protein content of the substrate was shown to increase the protein content of lesser mealworm [8]. The control substrate named B-Ad had a rather low content of protein (17.8%) compared to the high protein–high starch (HPHS) (32.7%) and high protein–low starch (HPLS) (39.1%) substrates [8]. The mealworms fed on control substrate had lower protein content but higher fat content than mealworms fed on the substrates HPHS and HPLS [8]. However, the high starch content of the substrate HPHS was observed to increase the fat content of mealworms [8]. Based on the study results of Van Broekhoven et al. [8], high protein content of substrate may cause a slight increase in the protein content of mealworms, but high protein in the substrate also improved survival and shortened the development time of this species.

### 2.5. Migratory Locust (Locusta migratoria L.)

In migratory locust, correlations between fat content and protein content, and between fat content and ash content, have been observed [35]. When the fat content increased, the protein and ash contents decreased [35]. An addition of wheat bran and carrots into the substrate increased the fat content and thus decreased the protein content in migratory locust [35]. In addition, contents of the minerals calcium, potassium, magnesium, and sodium of migratory locusts were significantly affected by the substrate [35]. It was also observed that the contents of Ca, K, Na, and P decreased in migratory locusts fed on the substrate containing wheat bran, although the wheat bran had a high concentration of P compared with the other substrates used [35]. On the other hand, the contents of Mg and Cu increased in insects due to high concentrations of these minerals in the wheat bran [35]. Iron content in migratory locusts was not observed to be affected by substrate [35]. In conclusion, the authors assumed that the nutritional composition of the migratory locust can be modified by the chemical composition of substrate [35].

## 3. The Effect of Substrate on the Fatty Acid Composition of Edible Insects

Linoleic (C18:2n6) and alpha linolenic (C18:3n3) acids are the most important fatty acids in the human diet because they cannot be synthesized in the body. Therefore, the human diet must contain certain amounts of linoleic and alpha linolenic acids. In Nordic nutrition recommendations, the amount of essential fatty acids is 3% of the daily energy intake, and one-sixth of the essential fatty acids are recommended to be alpha linolenic acid [36].

Edible insects often have low n-3 fatty acids content but a relatively high content of n-6 fatty acids [14], and thus an unfavorable n-6/n-3 ratio for human nutrition. Increasing the amount of omega-3 fatty acids in insects could also increase the nutritional value of insect products. In the following sections, the main findings regarding the fatty acid composition, saturated fatty acids (SFA), monounsaturated fatty acids (MUFA) and polyunsaturated fatty acids (PUFA) contents, n6 and n3 fatty acids, and essential fatty acids of edible insects and substrates are summarized. The contents of SFA, MUFA, PUFA, n-3 fatty acids, n-6 fatty acids, and the ratio of n-6 and n-3 in insects and in substrates used for rearing are presented in Table 1 and Table 2.

Our calculations were performed in order to evaluate how efficiently linoleic acid (C18:2n6) and alpha linoleic acid (C18:3n3) can be transferred from substrate to the insects. Spearman correlation test (IBM SPSS Statistics, Version 27, Amonk, NY, USA) was applied to obtain the correlations. Based on the calculations, it appears that substrate can affect the linoleic and alpha linolenic acids contents of insects (Table 3). When comparing correlation coefficients of linoleic and alpha linolenic acids between different insect species and rearing substrates, the correlation was, in fact, statistically significant (*p* < 0.01) for house cricket, BSF, and yellow mealworm.

Based on the studies addressed within this review, the amount of linoleic (C18:2n6) and alpha linolenic (C18:3n3) acids varies even in insects reared on similar control substrates [8,18,21,28,32,33,34,37]. This implies that in addition to the substrate used, other factors, such as the rearing environment, affect the amount of essential fatty acids in insects.

### 3.1. House Cricket (Acheta domesticus)

Linoleic acid (C18:2n6), oleic acid (C18:1n9), palmitic acid (C16:0), and stearic acid (C18:0) have been reported as the main fatty acids in house crickets [18,37]. Overall, concentrations of linoleic acid (C18:2n6) and alpha linolenic acid (C18:3n3) in house cricket varied based on the nutrient content of the substrate [18,37]. However, the exact same fatty acid was not detected in all cases in insects and in substrates, since some fatty acids may be synthesized or converted in insects to other fatty acids [18]. Only the concentration of palmitic acid (C16:0) in the house crickets was observed to reflect its content in the substrate [18].

The concentration of linoleic acid (C18:2n6) in house crickets fed on control substrate was 28.7–34.9% of TFA, and the concentration of alpha linolenic acid (C18:3n3) 0.8–1.2% [18,37]. Flaxseed oil, containing a notable amount of omega-3 fatty acids (57% alpha linolenic acid C18:3n3), was added to the substrate of house cricket in order to enhance the n-3 fatty acids content [37]. Flaxseed oil in substrate increased the crude fat content and also the concentrations of alpha linolenic acid (C18:3n3) and stearic acid (C18:0), and decreased the content of other fatty acids [37]. In addition, the authors observed that the fatty acid profile in house cricket reflected the fatty acid profile of the substrates, and an increase of one percentage unit of flaxseed oil in the substrate increased the alpha linolenic acid (C18:3n3) content in insects from 2.3% to 2.7% [37]. At the same time, the PUFA content increased and the ratio of n6/n3 decreased, and thus the nutritional quality of insects improved [37].

### 3.2. Black Soldier Fly (Hermetia illucens)

BSFs, both prepupae and larvae, were shown to contain mostly SFA (61.3–86.9%) of all fatty acids, although the majority of the used substrates contained mostly unsaturated fatty acids (UFA) [18,21,24,37]. The most common individual fatty acid in BSF prepupae or larvae reared on different substrates was lauric acid (C12:0) [18,21,24,28,37,38], although in some substrates the highest concentration of all the fatty acids was linoleic acid (C18:2n6) [21,24]. Linoleic acid (C18:2n6) in BSFs fed on control substrate was 8.1–11.6% of TFA or of total fatty acid methyl esters (FAME), and the concentration of alpha linolenic acid (C18:3n3) was 0.5–0.7% [18,21,28,37].

It seems that substrate with a high carbohydrate content increases SFA and especially lauric acid (C12:0) concentration in BSF prepupae [21,23,28]. BSF reared on substrate BRE were shown to have the lowest amount of SFA and the highest amount of polyunsaturated fatty acids (PUFA) compared to BSF reared on other tested substrates, even though BRE had the highest content of SFA [24]. BSF larvae reared on substrates BRE and WIN had the highest amounts of unsaturated linoleic acid (C18:2n6) and total PUFA compared to larvae reared on other substrates, but there were also large amounts of unsaturated oleic acid (C18:1n9) in all groups of larvae [24]. Furthermore, Galassi et al. [26] observed that substrate had an influence on the total amount of SFA in BSF larvae (*p* = 0.009), and that the substrate had a statistically significant effect on the lauric acid (C12:0) (*p* = 0.035) and myristic acid (C14:0) (*p* = 0.021) concentrations of larvae. Additionally, oleic acid (C18:1n9) concentration was seen to be affected significantly (*p* = 0.011) by the substrate [26].

According to Meneguz et al. [24], the fatty acid composition of BSF larvae is not affected only by fatty acid composition of the substrate but also by other nutritional components, such as carbohydrates. Oonincx et al. [18] observed that there was only a limited opportunity to modify fatty acid composition of BSF by substrate due to the fat metabolism of BSF. The substrates used contained a moderate amount of linoleic acid (C18:2n6), but after rearing, the amount of linoleic acid was quite low in BSFs [18]. They also observed that the fatty acid profile of BSF generally did not have the same trend as the fatty acid profile of the substrate [18]. Obviously, other components in the substrate, such as the amount of protein, affected the fat composition of BSF, although the concentration of palmitic acid (C16:0) was observed to be affected by the substrate [18].

Danieli et al. [21] observed that the fatty acid profile of BSF prepupae could be affected by the fatty acid content of substrate, and a positive correlation in UFA content was demonstrated between substrate and prepupae. In prepupae reared on substrate TMD1, with a higher concentration of SFA than the other tested substrates, higher SFA content was also observed, but a statistically significant (*p* < 0.01) positive correlation between SFA of substrate and prepupae was demonstrated only in the case of myristic acid (C14:0) [21]. The MUFA content of BSFs reared on TMD1 was lower than in BSFs reared on other substrates [21]. Moreover, lauric acid (C12:0) and myristic acid (C14:0) concentrations in BSF prepupae were observed to increase due to the high content of NFC in the substrate TMD1 [21].

The byproduct coffee silverskin (CS) was enriched with different amounts of microalgae, *Schizochytrium* sp. and *Isochrysis* sp., and was used as substrate for rearing BSF prepupae [39]. Inclusion of microalgae in the CS increased the total lipid content, decreasing the SFA content and increasing the UFA content, in substrate compared to CS without microalgae [39]. After rearing, a statistically significant positive correlation (*p* = 0.035) was observed in lipid content between substrate enriched with *Schizochytrium* sp. and prepupae [39]. However, statistically significant correlation was not observed in lipid content between substrate enriched with *Isochrysis* sp. and prepupae [39]. *Schizochytrium* sp. in CS caused a significant increase of docosahexaenoic acid (DHA, 22:6n3) and eicosapentaenoic acid (EPA, 20:5n3), and a decrease of SFA, in prepupae compared to CS without microalgae or CS with *Isochrysis* sp. [39]. A total of 10% of *Schizochytrium* sp. in CS was found to be sufficient to improve the nutritional quality of BSF by increasing the content of UFA, and especially the amount of n3 fatty acids [39].

**Table 1 insects-13-00590-t001:** Contents (% of total fatty acids) of saturated fatty acids (SFA), monounsaturated fatty acids (MUFA), and polyunsaturated fatty acids (PUFA) in feed substrates and in insects.

Common Name/ Scientific Name	Substrate Name	SFA		MUFA		PUFA		Reference
		Subst	Insect	Subst	Insect	Subst	Insect	
House cricket/ *Acheta domesticus*	Control, 0%	29.1	37.3	31.5	31.5	38.3	29.8	[37]
	Diet 1% FSO	24.4	37.0	29.2	30.6	45.5	31.0	
	Diet 2% FSO	22.6	34.6	28.5	30.4	48.0	33.7	
	Diet 4% FSO	18.7	31.9	25.9	28.4	54.9	38.4	
Lesser mealworm/ *Alphitotobius* *diaperinus*	Control, 0%	29.1	34.0	31.5	36.0	38.3	28.6	[37] ^1^
	Diet 1% FSO	24.4	31.2	29.2	35.6	45.5	31.9	
	Diet 2% FSO	22.6	30.7	28.5	34.5	48.0	33.6	
	Diet 4% FSO	18.7	31.0	25.9	32.5	54.9	35.2	
Black soldier fly/*Hermetia* *illucens*	C	27.31	79.58	21.72	10.55	50.96	9.87	[21] ^3^
	TMD1	35.11	86.89	14.15	8.49	50.74	4.62	
	TMD2	29.76	81.05	15.71	9.01	54.53	9.94	
	TMD3	29.33	81.35	15.33	8.87	55.34	9.78	
	VEGFRU	22.06	78.90	9.27	12.33	68.67	8.77	[24] ^2^
	FRU	24.80	81.88	24.42	13.34	50.78	4.78	
	WIN	15.37	63.01	19.80	18.97	64.83	18.02	
	BRE	27.19	61.25	11.25	12.74	61.56	26.01	
	Chicken feed	21.46	77.44	25.53	10.01			[28] ^3,4^
	Digestate	48.32	64.82	18.98	19.08			
	Vegetable waste	40.68	82.80	11.96	9.54			
	Restaurant waste	54.05	78.29	28.94	11.99			
	Control, 0%	29.1	74.4	31.5	15.1	38.3	10.1	[37] ^3^
	Diet 1% FSO	24.4	70.8	29.2	15.3	45.5	13.3	
	Diet 2% FSO	22.6	68.4	28.5	15.3	48.0	15.8	
	Diet 4% FSO	18.7	63.5	25.9	15.6	54.9	20.3	
Yellow mealworm/ *Tenebrio molitor*	WB	18.97	22.96	21.35	47.29	59.43	29.86	[32] ^4^
	BSG	25.08	24.38	12.70	24.35	62.21	51.38	
	S1	19.68	24.85	19.26	55.92	60.43	19.18	[33]
	S2	17.33	26.11	21.79	47.30	59.58	26.52	
	S3	19.14	23.67	36.49	58.09	43.80	18.24	
	S4	19.13	21.72	48.42	59.16	31.68	19.14	
	S5	17.16	20.37	59.98	60.56	22.32	19.08	
	MSCS	16.51	22.3	22.03	36.5	61	39.9	[34] ^4^
	HDSM	17.96	21.8	15.98	22.3	56	36.8	
	SDG	17.92	15.4	23.31	25.2	58.09	31.4	
	WB	19.52	25.3	16.75	33.8	61.8	38.5	
	D1	29.17	29.79 ^1^	32.04	45.62 ^1^	38.79	23.70 ^1^	[40] ^5^
			29.60 ^2^		50.01 ^2^		19.89 ^2^	
	D2	20.95	30.30 ^1^	41.44	46.67 ^1^	37.61	22.61 ^1^	
			29.33 ^2^		46.81 ^2^		23.58 ^2^	
	D3	19.63	30.63 ^1^	32.44	46.80 ^1^	47.93	22.01 ^1^	
			30.94 ^2^		48.26 ^2^		20.48 ^2^	
	D4	22.13	30.51 ^1^	36.83	47.11 ^1^	41.04	21.82 ^1^	
			31.99 ^2^		48.49 ^2^		19.26 ^2^	
	D5	22.59	30.26 ^1^	37.08	47.94 ^1^	40.33	21.19 ^1^	
			31.03 ^2^		47.92 ^2^		20.75 ^2^	
	D6	22.37	31.61 ^1^	35.86	49.96 ^1^	41.78	17.87 ^1^	
			32.06 ^2^		50.37 ^2^		17.33 ^2^	

^1^ Pupae, ^2^ larvae, ^3^ prepupae, ^4^ % of FAMEs (fatty acid methyl esters), ^5^ FA (mol%).

**Table 2 insects-13-00590-t002:** Contents (% of total fatty acids) of n-3 fatty acids and n-6 fatty acids and the ratio of n-6 and n-3 in feed substrates and in insects.

Common Name/ Scientific Name	Substrate Name	n-3		n-6		n-6/n-3		Reference
		Subst	Insect	Subst	Insect	Subst	Insect	
House cricket/ *Acheta domesticus*	Control					16.4	22.2	[18]
	HPHF					10.7	15.3	
	HPLF					4.9	29	
	Control, 0%	3.0	0.8	35.0	28.8	11.8	36.2	[37]
	Diet 1% FSO	14.2	4.1	31.1	26.8	2.2	6.6	
	Diet 2% FSO	22.2	7.2	25.6	26.4	1.2	3.7	
	Diet 4% FSO	30.5	12.7	24.2	25.6	0.8	2.0	
Lesser mealworm/ *Alphitotobius* *diaperinus*	Control B-Ad					12	19	[8] ^2^
	HPLS					18	19	
	LPHS					5	31	
	Control, 0%	3.0	1.2	35.0	27.0	11.8	21.7	[37] ^1^
	Diet 1% FSO	14.2	4.4	31.1	27.2	2.2	6.3	
	Diet 2% FSO	22.2	7.2	25.6	26.1	1.2	3.6	
	Diet 4% FSO	30.5	10.9	24.2	24.0	0.8	2.4	
Black soldier fly/*Hermetia* *illucens*	Control					20.1	15.1	[18] ^2^
	HPHF					10.7	11.1	
	HPLF					4.9	7.2	
	LPHF					13.5	9.1	
	LPLF					6.2	6.1	
	Chicken feed	2.85	0.86	50.10	11.59			[28] ^3,4^
	Digestate	7.11	1.60	17.56	8.04			
	Vegetable waste	14.97	2.33	31.93	4.62			
	Restaurant waste	2.18	1.43	14.24	8.00			
	Control, 0%	3.0	0.5	35.0	9.1	11.8	18.3	[37] ^3^
	Diet 1% FSO	14.2	3.3	31.1	9.7	2.2	3.0	
	Diet 2% FSO	22.2	5.5	25.6	10.0	1.2	1.8	
	Diet 4% FSO	30.5	9.7	24.2	10.4	0.8	1.1	
Yellow mealworm/ *Tenebrio molitor*	Control B-Tm/Za					10	19	[8]
	HPHS					16	32	
	HPLS					18	21	
	Control 1					11.1	26.6	[18]
	Control 2					13.5	45.2	
	HPHF					10.7	32.1	
	HPLF					4.9	102.1	
	LPHF					13.5	79.1	
	LPLF					6.2	40.6	
	WB	5.87	1.66	53.56	27.97			[32] ^4^
	BSG	7.02	3.64	55.19	47.57			
	S1					14.74	62.94	[33]
	S2					12.74	25.52	
	S3					13.85	42.77	
	S4					13.59	36.77	
	S5					14.43	39.37	
	MSCS	10	1.95	51	37.913	5.10	19.44	[34] ^4^
	HDSM	22.8	1.67	33.2	5.1229	1.46	21.03	
	SDG	8.59	1.86	49.5	0.57	5.76	15.90	
	WB	3.8	1.97	58	36.5	15.26	18.53	
	D1					21.55	32.09 ^1^	[40] ^5^
							35.64 ^2^	
	D2					34.27	36.11 ^1^	
							31.54 ^2^	
	D3					25.51	44.06 ^1^	
							40.53 ^2^	
	D4					29.54	38.17 ^1^	
							39.06 ^2^	
	D5					28.79	38.67 ^1^	
							34.68 ^2^	
	D6					26.91	41.09 ^1^	
							37.05 ^2^	

^1^ Pupae, ^2^ larvae, ^3^ prepupae, ^4^ % of FAMEs (fatty acid methyl esters), ^5^ FA (mol%).

**Table 3 insects-13-00590-t003:** Correlation coefficients (Spearman) between linoleic and alpha linolenic acid contents in house cricket (*Acheta domesticus*), black soldier fly (*Hermetia illucens*), yellow mealworm (*Tenebrio molitor* L.) and lesser mealworm (*Alphitobius diaperinus*), and the linoleic and alpha linolenic acid contents used in the rearing substrates.

	House Cricket	Black Soldier Fly	Yellow Mealworm	Lesser Mealworm
Linoleic acid (C18:2n6)	0.89 *	0.60 *	0.60 *	0.71
Alpha linolenic acid (C18:3n3)	0.94 *	0.59 *	0.66 *	0.75
References	[18,37]	[18,21,24,26,28,37,38]	[8,18,32,33,34,40]	[8,37]

* Correlation is significant at the 0.01 level (2-tailed).

Flaxseed oil, containing 57% alpha linolenic acid (C18:3n3), was added to the substrate of BSF [37]. The fatty acid profile of BSF reflected the fatty acid profile of the substrates, and an increase of one percentage unit of flaxseed oil in the substrate increased the alpha linolenic acid (C18:3n3) content from 2.3% to 2.7% [37]. In addition, the PUFA content increased and the ratio of n6/n3 decreased [37].

Fischer et al. [38] studied spent coffee grounds, donut dough, and their mix (1:1) as rearing substrates for BSF larvae. They found that with the blended mix of spent coffee and donut dough, an equal nutritional quality of larvae was achieved to that obtained using soybean meal [38]. Generally, fatty acids, especially UFA content, were higher in larvae fed on spent coffee, although total lipid content was higher in larvae fed on donut dough or blended mix [38]. SFA contents in larvae were the highest with blended substrate mix, although the substrate spent coffee had the highest SFA content [38]. Lauric acid (C12:0) was the most abundant fatty acid in larvae with all the substrates used, but it was significantly lower in larvae fed on spent coffee than in larvae fed on donut dough or blended mix [38]. By contrast, the most common individual fatty acids in the substrates spent coffee grounds, donut dough, and blended mix were linoleic acid (C18:2n6), oleic acid (C18:1n9), and palmitic acid (C16:0), respectively [38]. Overall, oleic acid (C18:1n9) and linoleic acid (C18:2n6) were shown to be the most common unsaturated fatty acids in BSF, regardless of the chemical composition of the substrate [18,21,24,28,37,38].

### 3.3. Yellow Mealworm (Tenebrio molitor L.) and Lesser Mealworm (Alphitobius diaperinus)

Palmitic acid (C16:0), oleic acid (C18:1n9), and linoleic acid (C18:2n6) have been shown to be the most prevalent fatty acids in yellow mealworm and in lesser mealworm [8,18,32,33,34,40]. According to Oonincx et al. [18], the fatty acid profile of yellow mealworm would appear to be rather constant, regardless of the different substrates. By contrast, Van Broekhoven et al. [8] showed that individual fatty acid composition, especially oleic and linoleic acids, varied based on the substrate, but the fatty acid profile in mealworms did not necessarily follow the same trend as the fatty acid profile in substrate. For example, a high amount of polyunsaturated C18 fatty acids in the substrate was observed to decrease the amount of C18 monounsaturated fatty acids in mealworms [8]. Alves et al. [9] observed that there were no statistically significant differences in fatty acid composition in yellow mealworms reared on control substrate A (containing 50% wheat flour and 50% soybean flour) or substrate B (containing 50% substrate A and 50% of bocaiuva pulp flour), but a small amount of caprylic acid (C8:0) was detected in mealworms fed on substrate B.

Melis et al. [32] reported that yellow mealworm larvae reared on substrate BSG had more PUFA, such as alpha linolenic acid (C18:3n3) and linoleic acid (C18:2n6), but less MUFA than mealworms reared on substrate WB. Thus, larvae fed on BSG accumulated alpha linolenic acid and linoleic acid more efficiently than larvae fed on WB [32]. By contrast, the MUFA content of larvae better reflected the amount of oleic acid (C18:1n9) in substrate [32]. Overall, according to Melis et al. [32], the fatty acid profile of yellow mealworm larvae can be modified by substrates.

In a study by Ruschioni et al. [33] of substrates containing olive pomace, the amount of MUFA was the highest in substrate S5, but PUFA were found most in substrate S3. Mealworm larvae reared on substrate S5 had the highest amount of MUFA, whereas the highest amount of PUFA was observed in larvae fed on substrate S4 containing 50% olive pomace [33]. Addition of olive pomace to the substrate decreased the amount of linoleic (C18:2n6) and alpha linolenic (C18:3n3) acids in substrate compared to substrates with no olive pomace, but no significant differences in the concentrations of linoleic acid or alpha linolenic acid in the mealworm larvae were observed, except in larvae fed on substrate S2 (middlings, containing no olive pomace) [33]. However, addition of olive pomace to the substrate did not change the fatty acid composition of yellow mealworms [33].

Dreassi et al. [40] reared yellow mealworms on six different substrates with increasing total fat content (0.46–9.34%). SFA, MUFA, and PUFA contents of the substrates were 19.6–29.2, 32.0–41.4, and 37.6–47.9 mol%, respectively [40]. After rearing, the crude fat content of larvae and pupae fed on the different substrates was rather stable, but the fatty acid profile in mealworms changed according to the substrate [40]. However, the fatty acid profile did not follow the profile of the substrate, although the most abundant fatty acids were the same in substrates and mealworms [40]. The lowest SFA content and the highest UFA content were in mealworms fed on the substrates 100% bread (D1) and 100% oat flour (D2) [40]. However, the substrates D1 and D2 had the lowest contents of PUFA, and the substrate D1 had also the lowest content of MUFA and the highest content of SFA [40].

The n6/n3 ratio was high in all yellow and lesser mealworms reared on substrates with different n6/n3 ratios, which was thought to be due to carrot included in the diet [8]. However, the results of a study by Oonincx et al. [18] indicated that the ratio of n6/n3 was higher in yellow mealworms fed on substrate without carrot than in mealworms fed on substrate with carrot. The ratio of n6/n3 varied in the range 15.9–21.0 in yellow mealworms fed on three byproducts, and the sum of UFA in mealworms was highly variable (56.6–76.4% of FAME) [34]. In a study by Oonincx et al. [18], yellow mealworm had the highest n6/n3 ratio compared to other insects, such as BSF and house cricket, and the ratio of n6/n3 in insects was not observed to follow the ratio of n6/n3 in substrates.

Lesser mealworms reared on control substrate had 22.7–26.9% linoleic acid (C18:2n6) and 0.7–1.2% alpha linolenic acid (C18:3n3) of TFA [8,37]. Wheat bran or wheat flour is usually used for rearing yellow mealworms, and as a control substrate in studies. Mealworms fed on control substrate had 18.8–36.5% linoleic acid (C18:2n6) and 0.3–2.0% alpha linolenic acid (C18:3n3) of TFA or of FAME [8,18,32,33,34]. Flaxseed oil, containing 57% alpha linolenic acid (C18:3n3), added to the substrate increased the crude fat content in lesser mealworms and also the concentrations of alpha linolenic acid (C18:3n3) and stearic acid (C18:0), and decreased the content of other fatty acids [37]. In addition, the fatty acid profile was observed to reflect the fatty acid profile of the substrates, and an increase of one percentage unit of flaxseed oil in the substrate increased the alpha linolenic acid (C18:3n3) content in insects from 2.3% to 2.7% [37]. At the same time, the PUFA content of insects increased and the ratio of n6/n3 decreased [37].

## 4. Discussion

Interest in edible insects has increased in Europe in recent years, and novel food applications for many insect species are pending in the EU. Approval processes for the applications are currently in progress, which requires active observation of changes of legislation. Edible insects could be a new sustainable protein source for human diet and animal feed. Thus, the quality and nutritional content of insects are important from a health aspect. However, the number of studies dealing with nutritional composition of edible insects and the effects of the used substrates is currently very limited, thus hampering extensive conclusions.

The chemical composition of insects is affected by many factors, such as species, substrates, and developmental stage of the insect [8,18]. For example, the nutrient composition of BSF varies at different stages of the life cycle, and highest crude protein content has been found in postmortem adult stage and crude fat content in 14-day-old larvae, but unsaturated fat content started to decrease in seven-day-old larvae and older [41]. *Brachystola magna* (Girard) grasshoppers have been reported to have the highest contents of protein, fat, calcium, and magnesium in the adult stage, whereas iron and zinc concentrations are the highest in the nymph stage [42].

It appears clear that substrate has an impact on the nutritional composition of insects, in some species more than in others. The nutritional value of house crickets can be modified via formulation of the substrate, and it has been reported that high protein content of the substrate could increase not only the protein content of crickets, but also the amounts of sodium, calcium, phosphorus, and potassium [17,19]. According to Van Broekhoven et al. [8], yellow and lesser mealworms were able to regulate their body protein content, regardless of the protein content of substrate, by increasing uric acid excretion, whereas fat content and fatty acid profile of mealworms were more affected by the substrate. Nevertheless, the fatty acid profile of the mealworms did not necessarily show a similar trend to that of the fatty acid profile of the substrate [8]. Furthermore, low nutritional quality of the substrate could lead to low fat content of the insect [8]. However, in the study by Zhang et al. [34], no clear correlation between the nutritional composition of substrates and yellow mealworm was observed. Morales-Ramos et al. [43] observed in their study that a macronutrient ratio of 0.23 protein: 0.06 lipid: 0.71 carbohydrate was optimal for growth of yellow mealworm, and the intake of carbohydrates was seen to increase the efficiency of growth, whereas the intake of neutral detergent fiber had a negative impact on growth.

The contents of protein and carbohydrate and their ratio in the substrate have been assumed to affect the contents of protein and fat and their ratio in BSF [21], and according to Liu et al. [25], the most suitable nutritional ratio of substrate for BSF larvae is 2:1:2 protein:fat:digestible carbohydrate. According to Danieli et al. [21], NFC in substrate still have a rather unknown influence on the chemical composition of insects, but the content of UFA in BSF prepupae has been reported to correlate positively with the UFA level of the substrate. In a study by Pimentel et al. [44], it was demonstrated that protein accumulation in larval fat bodies of BSF depends on the protein content of substrate and the protein intake of larvae, and, on the other hand, a low content of protein in substrate and a high content of sugar increased lipid accumulation in the fat bodies. Fischer et al. [38] observed that both the substrate and the development stage had a significant influence on protein and lipid contents of BSF. However, in a study by Danieli et al. [21], increased protein content of the substrate TMD3 was not reported to have any advantage for growth, yield, or chemical composition of BSF prepupae. Furthermore, Galassi et al. [26] observed that the substrate had no effect on the protein content of BSF larvae.

Insect farming should be economically viable, and therefore the nutritional composition of substrate must be suitable for each species in order to maximize the yield, and low-cost substrate raw materials should be favored. Byproducts are a sustainable option for rearing edible insects, and by using them also the costs of insect farming could be decreased. Substrate has an impact on larval performance and feed conversion efficiency, and feed conversion efficiency has been observed to be higher with high-protein substrate than with low-protein substrates [8]. In addition, according to Sorjonen et al. [45], higher protein content in byproduct-derived substrates increased the yield and improved the growth performance of house crickets. The usability of byproducts or side streams depends on the efficiency of the insects to eat and to convert them to body mass, and, for example, Argentinean cockroach (*Blaptica dubia* (Serville)) and BSF are able to utilize different kinds of substrates better than yellow mealworm or house cricket [18].

In order to increase the utilization of byproducts, one option could be a treatment with microorganisms. Somroo et al. [46] treated soybean curd residue (SCR) with *Lactobacillus buchneri* and observed that the treatment increased the protein and fat contents of BSF larvae compared to larvae reared on non-treated SCR or artificial feed (AF). However, when rearing mealworms or other insects for human consumption, the fatty acids quality, and especially the n6/n3 ratio of the substrate, must be taken into consideration [8]. Furthermore, EU legislation EU/2017/1017 obligates ensuring substrate quality for the rearing of edible insects in Europe, and thus the use of food waste is neither legal nor appropriate.

Finally, the nutrient content of insects varies considerably between species [47,48]. In addition, processing methods, such as freezing, heating, or drying, but also packing, storage time, and storage conditions, can affect the chemical composition of insects [4,49,50,51]. Therefore, analysis of the nutritional composition should be performed for the final consumption products. Furthermore, chitin content should be analyzed separately, so that the actual nutritional value of the insects could be estimated. For example, chitin contents of BSF prepupae varied from 5.6% to 9.6% in dry matter [21,28], depending on the nutrient composition of the substrate. In addition, the chitin content in insect meals was observed to correlate negatively with protein digestibility [52]. Thus, low chitin content in insect protein may improve digestibility and enhance the nutritional quality of the protein.

It appears that nutrient content of insects can be modified by using a substrate which has been planned carefully for each insect species individually, including both macro- and micronutrients. This is because the ingredients of the substrate can be metabolized in the insect to other compounds, for example, carbohydrates can be converted to fats. For example, according to Finke [53], the fatty acid composition, vitamin E concentrations, carotenoid content, and possibly some vitamin B concentrations in edible insects could be modified and improved by substrate optimization. Overall, optimizing the composition and structure of substrate, rearing conditions, and rearing time for each insect species might help to standardize the nutritional composition of edible insects. When considering the health aspects of insects, the fatty acid profile, and especially the limited amount of SFA and the ratio of n6/n3, must be taken into account.

## Data Availability

The data presented in this study are available in the article or Appendix A.

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
