# Peer review of "The Effect of Substrate on the Nutrient Content and Fatty Acid Composition of Edible Insects"

_insects, 2022, doi:10.3390/insects13070590_

Round 1

Reviewer 1 Report

Comments on the review

The types of edible insects that can be grown on an industrial scale attract a lot of attention of researchers due to the need to solve the issues of providing the population with food, and farm animals and aquaculture facilities with feed. There are not so many industrially grown insect species. The reviewed work is devoted to these types, which is undoubtedly relevant. Technological problems of growing insects as agricultural species are largely related to the characteristics of feed substrates for insects, which are determined by the peculiarities of their metabolism. The article under review concerns the possibility of influencing the composition of fatty acids of cultivated edible insects through targeted feed selection. The authors rightly point out that these insects are sensitive to changes in the external environment, such as the composition of the feed components of the substrate on which they develop, temperature, relative humidity, etc. At the same time, there are works on the effect of feed on the amino acid composition of insect biomass. The novelty and dignity of the authors in the generalization of literary data on the possibility of influencing the composition of fatty acids of insects also with the help of feed.

The literature has been used quite fully, although of course it would be possible to add, for example, such publications:

Tschirner M., Simon A. Influence of different growing substrates and processing on the nutrient composition of black soldier fly larvae destined for animal feed // Journal of Insects as Food and Feed. 2015. V. 1(4): 249-259. Wageningen Academic Publishers ISSN 2352-4588 online, DOI 10.3920/JIFF2014.0008.

However, in general, the work quite fully covers the issues of lipid characteristics of the insect species under discussion when grown on different substrates.

The tables do not require additional comments.

Reviewer 2 Report

New sustainably  protein sources for food and feed is important for human being. The edible insects are important potential sources of high quality protein for human being. In this manuscript, the authors reviewed the effects of substate on the nutritional value of edible insects, and introduced some of the examples of edible insects, such as, House cricket, Black soldier fly, Yellow mealworm, Lesser mealworm, and Migratory locust. This manuscript is interesting and can tell people that insects are also a good source of food nutrition, but they ignored the tussah and silkworm, which have been widely eaten, I think it can be accepted after they can supplement more comprehensive insects. 

Reviewer 3 Report

Due to population growth, the demand for food is increasing significantly. In recent years, edible insects have received extensive attention because of their environmental friendliness and sustainability. The nutritional composition of the substrate is an important factor for the efficient growth of edible insects, so this manuscript reviews the effect of substrate on the nutritional value of edible insects, especially the composition of fatty acids, and makes relevant analysis. The overall literature review of the manuscript is comprehensive, properly cited and clearly organized, which is a good quality and comprehensive review article.

Specific comments to prepare this manuscript for publishing are listed below.

1. in Section 2 (line 98), insects also contain carbohydrates, mainly in the form of chitin and glycogen. Consider adding carbohydrates to this section.

2. In Section 2, the influence of substrates on minerals is involved, but in sections 2.3 and 2.4, minerals are not involved.

3. In Section 2.1 (line 114-116), the protein rich (22%) substrate and low protein (16%) substrate should be clearly described. What is the substrate?

4. In lines 116-117, “Furthermore, the protein-rich substrate may decrease the fat content of the crickets, due to e.g. the digestibility of protein”. what is the relationship between reducing fat content and digestibility? Please make it clear.

5. Line 143, What is the abbreviation TMD3? In line 144, TMD1 is the same as above.

6. In lines 182-183, what is the full name of DM? Too far from the first interpretation of DM (line111). Moreover, dry matter (DM) can be read better without abbreviations.

7. In line 183, what are the full names of LPHF and LPLF? This is the first occurrence.

8. The conclusions of many references are listed in Section 2.3, but there is a lack of summary comments. It is suggested to add some summary content.

9. In Section 2.4, there is only one reference about lesser mealworm. It is suggested to add this content and then add some summary content. Section 2.5 is also modified accordingly.

10. There are many abbreviations in Tables 1 and 2. It is recommended to use the full name directly for better reading.

11. in Table 3, I don't understand that in the Spearman correlation test results, the substrates of different references in the same insect are the same? If the substrates are different, How do you calculate the values in Table 3? 
